# Association between Eruption Sequence of Posterior Teeth, Dental Crowding, Arch Dimensions, Incisor Inclination, and Skeletal Growth Pattern

**DOI:** 10.3390/children10040674

**Published:** 2023-04-01

**Authors:** Marta García-Gil, José Antonio Alarcón, Alberto Cacho, Rosa Yañez-Vico, Juan C. Palma-Fernández, Conchita Martin

**Affiliations:** 1Department of Orthodontics, Faculty of Odontology, University Complutense of Madrid, 28040 Madrid, Spainconchitamartin@odon.ucm.es (C.M.); 2BIOCRAN (Craniofacial Biology, Orthodontics and Dentofacial Orthopedics) Research Group, University Complutense of Madrid, 28040 Madrid, Spain; 3Faculty of Odontology, University of Granada, 18071 Granada, Spain

**Keywords:** dental crowding, eruption sequence, arch dimensions, incisor inclination, skeletal growth pattern

## Abstract

Background: We conducted research to investigate the effects of the eruption sequence of posterior teeth, arch dimensions, and incisor inclination on dental crowding. Material and Methods: A cross-sectional analytic study was performed on 100 patients (54 boys and 46 girls; mean ages: 11.69 and 11.16 years, respectively). Seq1 (canine-3-/second premolar-5-) or Seq2 (5/3) eruption sequences were recorded in maxilla, and Seq3 (canine-3-/first premolar-4-) or Seq4 (4/3) eruption sequences in mandible; tooth size, available space, tooth size-arch length discrepancy (TS-ALD), arch lengths, incisor inclination and distance, and skeletal relationship were noted. Results: The most common eruption sequences in the maxilla and mandible were Seq1 (50.6%), and Seq3 (52.1%), respectively. In the maxilla, posterior tooth sizes were larger in crowded cases. In the mandible, anterior and posterior tooth sizes were larger in crowded patients. No relationship between incisor variables and the maxillo-mandibular relationship and dental crowding was found. A negative correlation between inferior TS-ALD and the mandibular plane was found. Conclusions: Seq1 and Seq 2 in the maxilla and Seq 3 and Seq 4 in the mandible were equally prevalent. An eruption sequence of 3–5 in the maxilla and 3–4 in the mandible is more likely to cause crowding.

## 1. Introduction

The term dental crowding describes the discrepancy between the space available and the size of the teeth, which, if reduced, will result in dental crowding or rotations. [1,2]. Its etiology is multifactorial [1,2,3,4,5,6]. Its prevalence among growing children and teenagers is 56% [7], although it ranges from 31% to 96.6%, depending on the country [6]. This is strongly related to the change of the arch dimensions when transitioning from temporal to permanent dentition [8], the “tooth size-arch length discrepancy” (TS-ALD) [9,10], and, lastly, the need to manage the leeway space [11], which is around 2 mm per quadrant [12], overall, when a premature tooth loss occurs or when the second phase of dental eruption occurs [13]. 

The sequence of eruption of permanent teeth is of critical importance for optimal occlusion development. If not controlled, it can result in crowding or severe space problems. It has been reported that the development of a correct occlusion is influenced by the following factors: sequence of eruption and preservation of the leeway space [8,11,14,15], facial growth pattern [3,5,10], skeletal relation between the maxilla and the mandible [13,15], dental size and morphology [2,9,12], physiological dental mesialization [11], arch dimensions [1,9,10,11,13,14], apical bone size [3,11], perioral muscles [14], molar and incisor inclination [5,15,16], and dental position or inclination during eruption [11], among many others.

Crowding is the most prevalent malocclusion, being the most popular chief complaint [13]. In terms of functional, neuromuscular, and aesthetic issues, malocclusions are the third most prevalent oral health problem, according to the World Health Organization. Due to this, dental crowding should be noted as a real problem that affects both patients and orthodontists. Furthermore, it may be useful to study the relationship between various factors and TS-ALD in order to prevent future dental problems, improve quality of life, and possibly avoid aggressive treatments in the future if such factors are not considered.

Thus, the aim of this study is to determine whether dental crowding is associated with posterior tooth eruption sequences; examine the possible relationship between TS-ALD and arch dimensions, tooth size, incisor inclination, the relationship between the maxilla and mandible, and facial growth patterns; and, finally, determine if all the variables mentioned above have a relationship to a concrete eruption sequence of posterior teeth.

## 2. Materials and Methods

### 2.1. Study Design

A cross-sectional analytical study was performed at the Orthodontic Department, School of Dentistry, Complutense University of Madrid. The Ethics Committee of the Hospital Clinico San Carlos de Madrid approved the study (internal code 20/618-E_Tesis, approval date 19 October 2020), and all patients signed informed consent forms. The manuscript was written in accordance with the recommendations for reporting cross-sectional studies (STROBE) [17].

### 2.2. Sample Selection

Patients attending the above-mentioned department between October 2020 and October 2022 were consecutively recruited if they met the eligibility criteria. We eventually recruited 100 patients (54 boys and 46 girls; mean ages: 11.69 ± 1.6, 11.16 ± 1.26 years, respectively). 

The inclusion criteria for this study were patients in the second phase of the mixed dentition; patients with active eruption of canines and second premolars in the maxilla and/or canines and first premolars in the mandible; clinical history; initial dental casts; and an initial lateral cephalogram. As part of the exclusion criteria, patients with the following conditions were excluded: complete permanent dentition, first phase of mixed dentition, or deciduous dentition, extraction of any primary teeth (due to caries or any other reason), previous orthodontic treatment, dental agenesis and craniofacial anomalies, systemic diseases, or chronic pharmacological treatment. 

Sample size was calculated using the following formula for cross-sectional studies [18]: n = (Z^2^ P (1 – P))/d^2^, where Z represents the statistic corresponding to level of confidence, P (represents the expected prevalence, and d represents precision (corresponding to effect size). Based on a confidence level of 95%, a prevalence of crowding of 56%, and a precision of d = 0.10, the resulting sample size was 91. In anticipation of possible dropouts, we rounded up the number of subjects to 100.

### 2.3. Outcomes

Among the data collected (Table 1) are the following.

The dental models were measured using a carbon fiber composite electronic digital caliper with 0.1 mm resolution and +/−0.2 mm accuracy, and cephalometric radiographs were analyzed using the software Dolphin Imaging 11.7 (Chatsworth, CA, USA).

Dental crowding was measured as TS-ALD (mm) by subtracting the space needed for eruption of the ten anterior teeth (mesio-distal tooth widths) from the space available in the dental arch between molars. The mesio-distal dimensions of the mandibular and maxillary teeth (generally at the anatomical contact point) were measured (Figure 1A). Unerrupted permanent canines and premolars were analyzed using Moyers mixed dentition space analysis.

The available space was measured by dividing each dental arch into four segments: two posterior (from the mesial surface of the canine/distal surface of permanent lateral incisor to the mesial surface of the first molar), one on each side, and two anterior (from the mesial surface of the canine to the midline). All segments were then compiled, as described by Sharma and Brown [19] (Figure 1B).

TS-ALD was further analyzed and categorized into two categories: positive (or zero) TS-ALD and negative TS-ALD.

The upper and lower sequences of eruptions (Seq) were the main independent variables. Sequences 1 and 2 were defined for the upper arch, and Sequences 3 and 4 were defined for the lower arch, as follows: Seq 1: maxillary canine (3) erupts in first place, and maxillary second premolar (5) erupts afterward; Seq 2: maxillary second premolar (5) erupts in first place, and maxillary canine (3) erupts afterward; Seq 3: mandibular canine (3) erupts in first place, and mandibular first premolar (4) afterward; Seq 4: mandibular first premolar (4) erupts in first place, and mandibular canine (3) erupts afterward. Sequences 2 in the upper arch and 3 in the lower arch are considered the most advantageous. Additionally, different sequences within the same dental arch (Seq 0) were noted and reported as asymmetrical cases.

Using a straight line between the interdental papilla tip between the central incisors and a tangent through the distal surfaces of the first molars, we measured the total upper and lower arch lengths. The total arch length is comprised of two components: anterior (from the papilla to the canines’ cusps) and posterior (from the canines to molars) (Figure 1C).

Maxillary and mandibular intercanine arch widths were measured from the cusp tip of canine on one side to the cusp tip of canine on the opposite side of the same arch. Intermolar arch widths for the maxilla and mandible were calculated from the central fossa of the first permanent molar on one side to the central fossa of the first permanent molar on the other side (Figure 1D).

### 2.4. Statistical Analysis

Descriptive statistics used means and standard deviations (SD), along with 95% confidence intervals (CI) and frequencies and proportions, depending on the type of variable (quantitative or qualitative).

Chi-square test was used to compare the sequence of eruption for each arch (upper and lower) with different outcomes. The Shapiro–Wilk test was used to confirm the normality of the variable distribution. The Student T-test was then used to compare the results according to the sequences, and the Pearson correlation coefficient was used to determine if crowding and cephalometric measurements were correlated.

The replicability of the measurements was evaluated with the intraclass correlation coefficient (ICC) by comparing repeated measures from 10 patients measured on five different days.

The statistical analysis was performed using SPSS version 22 (SPSS Inc., Chicago, IL, USA), and the level of significance was set at *p* ≤ 0.05.

## 3. Results

### 3.1. Demographic Characteristics of the Sample

The final sample included 100 patients, 46 females, and 54 males. The average age was 11.45 ± 1.47 years, ranging from 7.8 to 16.74. The Chi-square test was non-significative for sex (*p* = 0.64), indicating that the sample was homogeneous for this variable. Eighty-one upper arches and forty-eight lower arches fulfilled the eligibility criteria.

### 3.2. Sequence of Eruption

Maxillary sequences: the prevalence of Seq1 (3–5) was 50.6%, whereas for Seq2 (5–3), it was 43.2%. 6.1% were asymmetrical cases. Regarding gender differences, 27.1% of men and 23.5% of women showed Seq1, while 22.2 of men and 21.1% of women presented Seq2. In addition, 4.9% of men and 1.2 % of women had asymmetrical sequences (Figure 2). Differences between sexes were not statistically significant (*p* = 0.689).
Figure 2Sex-related frequencies of upper eruption sequences.
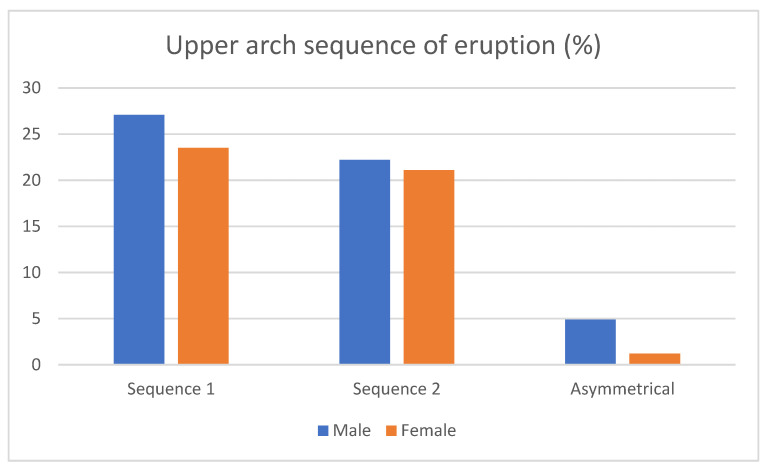



Mandibular sequences: the prevalence of Seq3 (3_4) was 52.1%, whereas for Seq4 (4_3), it was 43.7%. Additionally, 4.2% of the patients showed asymmetrical sequences in the lower arch. On the one hand, 25% of men and 27.1% of women showed Seq3, and 33.3 % of men and 10.4% of women presented Seq4, leaving an equal distribution of asymmetrical patients (2.1 of men and 2.1 of women) (Figure 3). We found no difference between sexes for the mandibular sequences according to the Chi-square test (*p* = 0.153).
Figure 3Sex-related frequencies of lower eruption sequences.
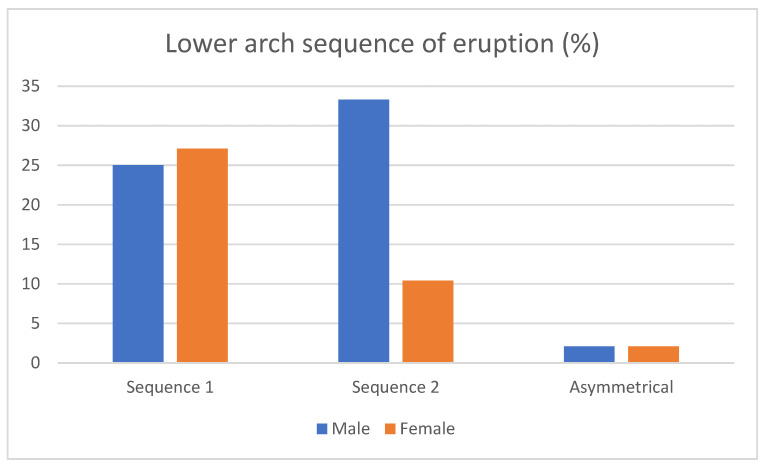



Analysis of eruption sequences, TS-ALD, arch dimensions, sagittal and vertical growth patterns, and incisor position and inclination.

Maxillary sequences: Table 2 shows the comparisons for different outcomes between children with eruption sequences 1 and 2. The *t*-test showed non-significant differences for all outcomes. Intercanine width was slightly larger in the Seq 1 children (mean: 33.9 mm; SD: 2.85) than in Seq 2 (mean: 32.25 mm; SD: 2.82) (*p* = 0.052, almost significant).
children-10-00674-t002_Table 2Table 2Comparisons between upper eruption sequences, TS-ALD (tooth size- arch length discrepancy), arch dimensions, sagittal and vertical growth patterns, and incisor position and inclination. SD: standard deviation.Maxillary ArchSequence 1Sequence 2Mean Difference (Diff.)95% Confidence Interval of the Mean Diff.*p* Value

MeanSDMeanSD
LowerUpper
Age (years)
11.561.2511.291.420.28−0.330.890.369TS-ALD (mm)
−1.164.63−1.884.750.72−1.452.880.511Tooh size_5_4_3 (right) (mm)22.221.8321.992.100.23−0.671.120.618Tooh size_21_12 (mm)31.152.4531.052.730.10−1.091.280.871Tooh size_3_4_5 (left) (mm)22.482.3021.892.080.59−0.421.600.251Total arch space (mm)74.504.3273.114.901.38−0.723.490.195Space_5_4_3 (right) (mm)21.942.2021.162.490.78−0.291.850.152Space_21_12 (mm)31.432.1431.962.30−0.53−1.550.480.298Space_3_4_5 (left) (mm)21.312.3120.682.550.63−0.481.740.262Anterior arch length (mm)14.071.9613.972.570.11−0.931.140.837Posterior arch length (mm)25.832.7325.311.350.51−0.501.530.314Total arch length (mm)39.893.1239.282.840.61−0.761.980.378Intermolar width (mm)45.642.5444.743.080.90−0.392.180.169Intercanine width (mm)33.902.8532.252.821.65−0.023.320.052Inclination upper incisor_NA (°)24.896.3922.246.682.65−0.355.640.082Distance upper incisor_NA (mm)5.522.364.472.381.05−0.042.130.059ANB angle (°)
2.962.493.862.23−0.89−1.980.200.106Witts (mm)
−1.562.99−0.803.30−0.76−2.200.680.297Mandibular plane angle(°)25.247.2224.494.210.76−2.003.520.586Lower facial height (°)44.445.2745.213.73−0.78−2.901.350.469


Maxillary eruption sequences and categorized TS-ALD: A Chi-square test was used to compare the two upper eruption sequences between children with negative TS-ALD and children with positive or zero TS-ALD. No significant differences were found (*p* = 0.161).

Results showed that 19.8% of patients with Seq1 had positive or zero TS-ALD, and 30.9% showed negative TS-ALD. In addition, 16% of patients with Seq2 showed positive or zero TS-ALD and 27.2% exhibited some degree of crowding. Regarding patients with asymmetrical sequences, only 1.2% had space problems, while 4.9% presented zero or positive TS-ALD. In both Seq1 and Seq2, dental crowding was more frequent, being even more usual in Seq1 (Figure 4), but the differences were non-significant. 

Mandibular sequences: Table 3 shows the comparisons for different outcomes between children with eruption sequences 3 and 4. The t-test showed non-significant differences for all outcomes.
children-10-00674-t003_Table 3Table 3Comparisons between lower eruption sequences, TS-ALD (tooth size- arch length discrepancy), arch dimensions, sagittal and vertical growth patterns, and incisor position and inclination. SD: standard deviation.Mandibular ArchSequence 3Sequence 4Mean Difference (Diff.)95% Confidence Interval of the Mean Diff.*p* ValueMeanSDMeanSDLowerUpperAge (years)11.401.4110.761.600.65−0.251.540.152TS-ALD (mm)−0.154.430.223.34−0.37−2.742.000.754Tooh size_5_4_3 (right) (mm)21.962.0421.981.01−0.01−1.000.970.98Tooh size_21_12 (mm)23.151.3822.652.720.50−0.751.760.421Tooh size_3_4_5 (left) (mm)21.962.0721.871.090.10−0.921.110.847Total arch space (mm)66.803.3366.724.080.07−2.132.270.948Space_5_4_3 (right) (mm)22.041.7321.972.610.06−1.231.360.921Space_21_12 (mm)22.321.2522.411.66−0.09−0.960.770.829Space_3_4_5 (left) (mm)22.601.6522.102.090.50−0.611.610.365Anterior arch length (mm)8.702.078.922.27−0.23−1.521.060.724Posterior arch length (mm)27.031.9626.362.290.67−0.591.930.29Total arch length (mm)35.722.5435.282.380.45−1.021.920.543Intermolar width (mm)37.256.7538.324.76−1.07−4.532.400.538Intercanine width (mm)25.631.5225.461.990.17−1.061.390.784Inclination lower incisor_NB (°)24.965.0925.294.64−0.33−3.242.590.823Distance lower incisor_NB(mm)5.202.224.641.930.56−0.691.810.373ANB angle (°)3.422.004.311.67−0.89−2.000.220.113Witts (mm)−1.543.29−0.482.72−1.06−2.880.750.244Mandibular plane angle(°)27.408.0724.524.552.88−1.126.880.154Lower facial height (°)46.306.1044.673.941.63−1.494.750.297


Mandibular sequence and categorized TS-ALD: In order to compare the two lower eruption sequences between children with negative TS-ALD and children with positive or zero TS-ALD, a Chi-square test was used. There were no significant differences (*p* = 0.196).

Results showed TS-ALD was positive or zero in 20.8% of patients with Seq 3 and negative in 31.3%. Additionally, 29.2% of patients with Seq 4 showed positive or zero TS-ALD and 14.6% had some degree of crowding. There was no difference in results between both groups of TS-ALD patients with asymmetrical sequences. 

In Seq3, there appears to be a greater tendency to find a lack of space, but the differences were non-significant. (Figure 5).

### 3.3. Gender-Specific Analysis of TS-ALD, Arch Dimensions, Sagittal and Vertical Growth Patterns, and Incisor Position and Inclination

Despite no statistically significant differences between sexes for maxillary (*p* = 0.689) and mandibular eruption sequences (*p* = 0.153), comparisons between sexes were made for the remaining outcomes. Total upper arch space was larger in males (mean: 75.43 mm; SD: 4.30) than in females (mean: 72.54 mm; SD: 4.83) (*p* = 0.006). Total lower arch space was larger in males (mean: 68.23 mm; SD: 3.59) than in females (mean: 64.13 mm; SD: 4.51) (*p* = 0.001) too. 

The upper anterior space (male’s mean: 32.3 mm; SD: 2.29 vs. female’s mean: 30.94; SD:1.97), upper posterior left space (male’s mean: 21.74 mm; SD: 2.13 vs. female’s mean: 20.43; SD: 2.59), upper anterior tooth size (male’s mean: 31.67 mm; SD: 2.31 vs. female’s mean: 30.45; SD: 2.73), and upper posterior left tooth size (male’s mean: 22.66 mm; SD: 2.38 vs. female’s mean: 21.67; SD: 1.78), were also significantly larger in males (*p* = 0.006, *p*= 0.014, *p* = 0.033, and *p* = 0.041, respectively). However, upper TS-ALD was not significantly different between males and females. 

The lower posterior right space (male’s mean: 22.65 mm; SD: 2.15 vs. female’s mean: 21.33; SD: 2.06; *p* = 0.04) and left space (male’s mean: 23.06 mm; SD: 1.75 vs. female’s mean: 21.47; SD: 1.66; *p* = 0.003) were larger in males than females. Smaller tooth sizes and lower TS-ALD were similar between genders. 

Total upper arch length (male’s mean: 40.62 mm; SD: 2.87 vs. female’s mean: 38.73; SD: 2.83; *p* = 0.001) and anterior upper arch length (male’s mean: 14.86 mm; SD: 2.07 vs. female’s mean: 13.25; SD: 2.11) (*p* = 0.001) were larger in males than females. Posterior lower arch length (male’s mean: 27.33 mm; SD: 1.72 vs. female’s mean: 26.08; SD: 2.58; *p* = 0.05) was larger in males too.

Upper and lower intermolar widths were larger in males than females (upper male’s mean: 46.03 mm; SD: 2.54 vs. female’s mean: 44.42; SD: 2.93; *p* = 0.009; lower male’s mean: 39.34 mm; SD: 4.66 vs. female’s mean: 35.38; SD: 6.71; *p* = 0.021), respectively.

All cephalometric measures related to incisors inclination and position and skeletal relationships were similar between males and females, except for the mandibular plane angle, which was steeper in males (male’s mean: 26.39°; SD: 6.13 vs. female’s mean: 23.55; SD: 4.97; *p* = 0.014).

Males and females did not differ significantly in age.

### 3.4. Bivariate Correlations (Pearson’s Correlation Coefficient)

Continuous variables were correlated using the Pearson correlation coefficient. The main outcome of this analysis was dental crowding (TS-ALD). The conceptually related variables showed significant correlations, as expected.

Maxillary arch: we found a positive and moderate correlation between TS-ALD and available space (r = 0.373; *p* < 0.01), available posterior space (right side: r = 0.566; *p* < 0.001; left side: r = 0.470; *p* < 0.001), total arch length (r = 0.241; *p* = 0.031), anterior arch length (r = 0.315; *p* = 0.004), and intermolar width (r = 0.325; *p* = 0.003). The variables regarding dental size showed a negative correlation (r = −0.450; *p* < 0.001 for right posterior teeth; r = −0.362; *p* = 0.001 for left posterior teeth, and r = −0.281; *p* = 0.012 for anterior teeth). See Appendix A.

Mandibular arch: we found a positive and moderate correlation between TS-ALD and available posterior space (right side: r = 0.419; *p* = 0.003; left side: r = 0.380; *p* = 0.008) and a negative correlation with dental size (r = −0.554; *p* < 0.001 for right posterior teeth; r = −0.555; *p* < 0.001 for left posterior teeth, and r = −0.293; *p* = 0.043 for anterior teeth)) and the mandibular plane (r = −0.287; *p* = 0.048). See Appendix A. 

## 4. Discussion

This study investigates how the eruption sequence of posterior teeth, arch dimensions, incisor inclination, and skeletal growth pattern may affect dental crowding. Moreover, we sought to determine whether demographic factors, such as gender, influenced these sequences.

### 4.1. Sequence of Eruption

Our study examined dental eruption separately in the maxilla and mandible. In the maxilla, the most prevalent sequence was Seq1 (3-5), followed by Seq2 (5-3). Negative TS-ALD was more frequent in the maxilla than in diastemas, and it occurred more often when the eruption sequence was canine-second premolar (Seq 1). A negative TS-ALD was more common in the mandible when the eruption sequence canine-first premolar (Seq 3) occurred, which was the most common. 

Additionally, we compared all variables with the eruption sequence. Maxillary variables showed a nearly statistically significant difference in intercanine width between Seq 1 (33.9 mm) and Seq 2 (32.25 mm). Mandibular variables did not show any significant differences. The lack of differences between sequences in available space was an unexpected finding.

Moshkelgosha et al. [20] analyzed the relationship between the intrabony position of posterolateral teeth and dental crowding. They found that the 4-3-5 sequence was related to a significant lack of space. Padma Kumari et al. [21] found more space loss in patients where temporal canines experienced premature exfoliation when compared to a control group. 

### 4.2. Gender

A comparison of sex revealed that males showed specific larger spaces, arch lengths, tooth sizes, lower posterior arches, and upper and lower intermolar widths, as well as a steeper mandibular plane when compared to females. As a result, we could observe sexual dimorphism in our study. In spite of this, and contrary to Höffding [22] and Savara [23], we found that eruption sequences did not differ by gender, as other studies have reported [24]. It is possible that globalization and subsequent changes in eating habits have contributed to this.

### 4.3. Arch Dimensions

In our study, we found different results regarding the dimensions of the maxillary and mandibular arches. We did not find any differences between children with eruption sequences Sq1 and Sq2, nor did we find any differences between children with negative TS-ALD and positive or zero TS-ALD.

According to the definition of TS-ALD, correlation analysis showed a positive correlation between TS-ALD and available space, available posterior space, total and anterior arch length, and intermolar width. Instead, TS-ALD was negatively correlated with both anterior and posterior dental size.

In the mandibular arch, we found no difference between children with eruption sequences Sq3 and Sq4, nor between children with negative TS-ALD and positive or zero TS-ALD. Overall, our correlations showed a positive correlation between TS-ALD and right and left available space and a negative correlation between TS-ALD and dental size.

Based on their study, Janson et al. [1] found that arch length was negatively correlated with dental crowding. As for intercanine and intermolar widths, Selmani et al. [25], as well as Abid et al. [11], reported significantly narrower arch widths in patients with crowding. In their study, Faruqui et al. [2] found no statistical differences in intercanine widths between crowded and non-crowded arches; meanwhile, Ikoma et al. [26] observed that intercanine widths were wider, whereas intermolar widths were not different. According to Faruqui et al. [2], Bishara et al. [10], and Turkkahraman et al. [27], dental crowding was associated with decreased arch length.

### 4.4. Incisors Inclination and Position

The present study found no significant differences in incisor position or inclination between Seq 1 and 2 in the maxilla and Seq 3 and 4 in the mandible when compared to other studies, where incisor position and inclination were associated with lower anterior dental crowding [20,28,29,30,31]. 

### 4.5. Sagittal Skeletal Relationship

Neither eruption sequence nor dental crowding were correlated with skeletal variables. Conversely, Fernández 25 et al. [28] found increased Wits values (skeletal Class II) related to negative TS-ALD.

### 4.6. Vertical Growth Pattern

Our study found a negative correlation between lower TS-ALD and mandibular plane angle, indicating more crowding when the mandibular plane angle was increased. 

Lupacheva et al. [32] and Ikoma et al. [26] found that there was a relationship between vertical growth patterns and dental crowding, which is in accordance with the results of our study. 

In light of previous results as well as the characteristics of this study, a prospective longitudinal study is the next step to obtain more sustainable conclusions based on the previous results. In order to analyze the possible relationship between the variables given and the order of eruption more effectively, a prospective longitudinal study should be conducted to follow the natural development of the eruption of teeth. The timing of the eruption of the second permanent molar should also be investigated. 

## 5. Conclusions

In our study, eruption sequences 1 and 2 in the maxilla and 3 and 4 in the mandible were equally prevalent.Males and females showed similar eruption sequences both in the maxilla and in the mandible.Maxillary crowding occurred more frequently during eruption sequences 3-5 (Seq1), but the lack of space was evident in both maxillary sequences. There was more crowding in the mandible when the eruption sequence was 3-4 (Seq3), whereas diastemas and no space problems were prevalent when the eruption sequence was 4-3 (Seq4).

## Figures and Tables

**Figure 1 children-10-00674-f001:**
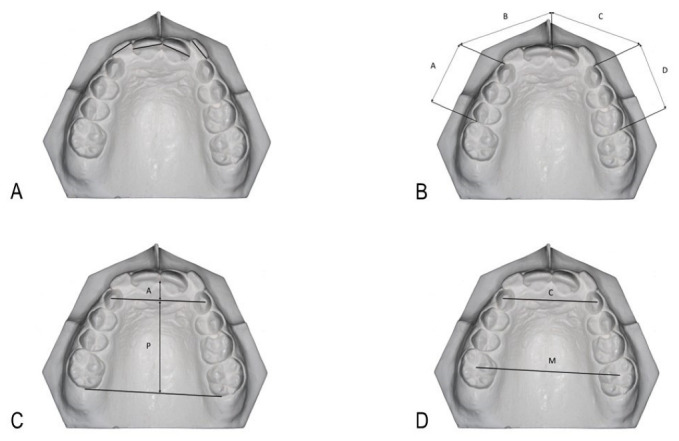
Measures taken in the dental models. (**A**): incisors tooth size; (**B**): available space in the arch, as described by Sharma and Brown [19]: upper_right canines and premolars (A), upper right and left incisors (B + C), upper_left canines and premolars (D). (**C**): Arch length, anterior (A), posterior (P), and total (A + P) arch length; (**D**): Intercanine (C) and intermolar (M) widths.

**Figure 4 children-10-00674-f004:**
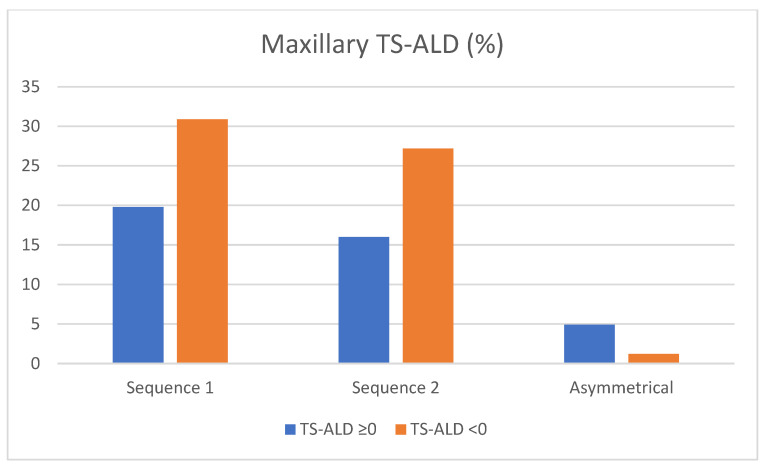
TS-ALD—related frequencies (percentages) of upper eruption sequences. TS-ALD < 0: negative TS-ALD (mm); TS-ALD ≥ 0: zero or positive TS-ALD (mm).

**Figure 5 children-10-00674-f005:**
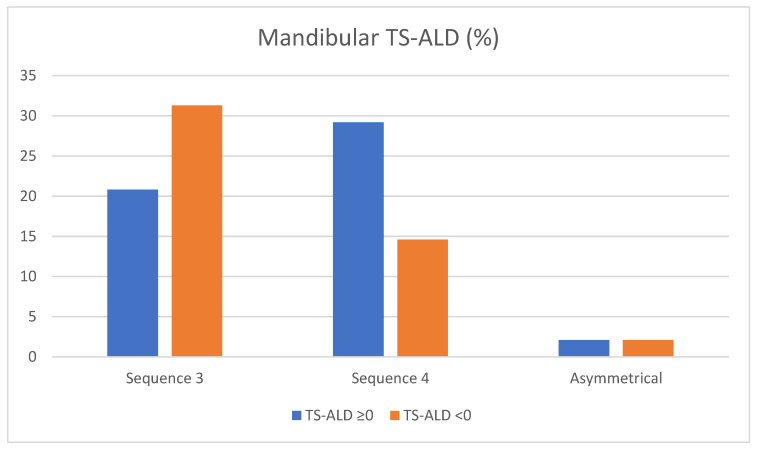
TS-ALD—related frequencies (percentages) of lower eruption sequences. TS-ALD < 0: negative TS-ALD (mm); TS-ALD ≥ 0: zero or positive TS-ALD (mm).

**Table 1 children-10-00674-t001:** Table 1 includes the main variables of this study. Sequence of eruption: Sequence 1: maxillary canine (3) erupts in first place, and maxillary second premolar (5) erupts afterward; Seq2 (Sequence 2): maxillary second premolar (5) erupts in first place, and maxillary canine (3) erupts afterward; Seq3 (Sequence 3): mandibular canine (3) erupts in first place, and mandibular first premolar (4) afterward; Seq4 (Sequence 4): mandibular first premolar (4) erupts in first place, and mandibular canine (3) erupts afterward; 0: asymmetry, with Seq1 and Seq2 or Seq3 and Seq4 in the same dental arch. Tooth size: upper_21_12: dental size of upper anterior teeth; upper_5_4_3: dental size of right upper posterior teeth; upper_3_4_5: dental size of left upper posterior teeth; lower_21_12: dental size of anteroinferior teeth; lower_5_4_3: dental size of right posteroinferior teeth; lower_3_4_5: dental size of left posteroinferior teeth. Available space: upper_21_12: space in upper anterior arch; upper_5_4_3: space in superior right-side arch; upper_3_4_5: space in upper left-side arch; lower_21_12: space in lower anterior arch; lower_5_4_3: space in lower right-side arch; lower_3_4_5: space in lower left-side arch. LFH: lower facial height.

Sequence (Seq) of eruption	Seq1 Seq2 Seq3 Seq4 0
Sex	0: men 1: women
Age (years)	
TS-ALD (tooth size—arch length discrepancy) (mm)	0 or positive: diastemas Negative: crowding
Tooth size (mm)	upper_21_12 (four incisors) upper_5_4_3 (right canines and premolars) upper_3_4_5 (left canines and premolars) lower_21_12 (four incisors) lower_5_4_3 (right canines and premolars) lower_3_4_5 (left canines and premolars)
Available space in the arch (mm)	upper_21_12 (four incisors) upper_5_4_3 (right canines and premolars) upper_3_4_5 (left canines and premolars) lower_21_12 (four incisors) lower_5_4_3 (right canines and premolars) lower_3_4_5 (left canines and premolars)
Arch length (mm)	Anterior arch length: upper and lower Posterior arch length: upper and lower Total arch length: upper and lower
Intermolar width (mm)	Upper Lower
Intercanine width (mm)	Upper Lower
Incisor inclination (°) (according to Steiner cephalometric analysis)	Upper incisor_NA line Lower incisor_NB line Interincisal angle
Incisor distance (mm) (according to Steiner cephalometric analysis)	Upper incisor_NA line Lower incisor_NB line
Sagittal skeletal relationship (from Steiner and Wits cephalometric analysis)	ANB angle (°) Wits (mm)
Growth pattern (°) (from Ricketts cephalometric studies)	Mandibular plane angle LFH

## Data Availability

The data presented in this study are available on request from the corresponding author.

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
