# Peer review of "Association between Eruption Sequence of Posterior Teeth, Dental Crowding, Arch Dimensions, Incisor Inclination, and Skeletal Growth Pattern"

_children, 2023, doi:10.3390/children10040674_

Round 1

Reviewer 1 Report

Dear Authors, congratulations on this study. It is very well presented and discussed the matter of timing and pattern of the eruption. 

As I like to add limitations to the study, although you did mention that a prospective study would be good, I think that you should add the limitation as you did not mention the reasons for shedding the primary teeth in this sample (spontaneous or extraction due to caries). And also that the timing of the eruption of the second permanent molar is interesting to investigate in relation to the seq. you investigated in this study. 

Great job.

Author Response

Reviewer: 1

Referee’s comment: Dear Authors, congratulations on this study. It is very well presented and discussed the matter of timing and pattern of the eruption.

As I like to add limitations to the study, although you did mention that a prospective study would be good, I think that you should add the limitation as you did not mention the reasons for shedding the primary teeth in this sample (spontaneous or extraction due to caries). And also that the timing of the eruption of the second permanent molar is interesting to investigate in relation to the seq. you investigated in this study.

Great job.

Author’s response: We thank the reviewer for his/her kind words and appreciate the reviewer's constructive comments. The suggestions made have been addressed in the Material and Methods and in the Discussion as follows:

Material and Methods: ….. “As part of the exclusion criteria, patients with the following conditions were excluded: complete permanent dentition, first phase of mixed dentition, or deciduous dentition, extraction of any primary teeth (due to caries or any other reason), previous orthodontic treatment, dental agenesis and craniofacial anomalies, systemic diseases, or chronic pharmacological treatment”

Discussion: “In order to analyze the possible relationship between the variables given and the order of eruption more effectively, a prospective longitudinal study should be conducted to follow the natural development of the eruption of teeth. The timing of eruption of the second permanent molar should also be investigated”.

Reviewer 2 Report

Dear author, I congratulate you on your work, I recommend:

- Improve the presentation of the results, there is content that is repeated in the text and in the tables, so I would summarize without repeating much information, highlighting what is relevant.

- On Page 7 square the table and the subtitle is cut at the page change.

Kind regards

Author Response

Reviewer: 2

Referee’s comment 1: Dear author, I congratulate you on your work, I recommend:

- Improve the presentation of the results, there is content that is repeated in the text and in the tables, so I would summarize without repeating much information, highlighting what is relevant.

Author’s response 1: Thanks for your suggestion to improve this section. The Results section has been revised.

Referee’s comment 2:

- On Page 7 square the table and the subtitle is cut at the page change.

Author’s response 2: Sorry for this editing mistake. We hope it has been solved. 

Reviewer 3 Report

Overall Rating

The purposes of this study are very interesting. However, clear and novel findings have not been obtained in the study. The authors need to reexamine their study. My minor comments are listed below:

1.        The authors should add the description of how the sample size was estimated in the Study settings and design section.

2.        Please illustrate the measurement points using schematic diagrams or photographs in dental model analysis in Methods section.

3.        Authors should provide a table of correlation coefficients for the measures.

4.        Although the aim of author’s study was to determine whether dental crowding is associated with posterior tooth eruption sequences, authors described that the t-test showed non-significant different for all outcomes in Results section. Therefore, authors should reconsider the purpose of the study (e.g. factors associated with crowding) and add items to be statistically processed.

Author Response

Reviewer: 3

Referee’s comment 1: Overall Rating: The purposes of this study are very interesting. However, clear and novel findings have not been obtained in the study. The authors need to reexamine their study. My minor comments are listed below:

Author’s response 1: We thank the reviewer for his/her suggestion. We hope the manuscript has improved after his/her comments.

Referee’s comment 2: The authors should add the description of how the sample size was estimated in the Study settings and design section.

Author’s response 2: Thank you for mentioning this point. According to our statistician, the following formula was used for sample size calculation in cross-sectional studies (see reference by Pourhoseingholi et al, 2013):

n= (Z2 P (1-P)) /d2, where Z is the statistic corresponding to level of confidence (95%, Z=1.96), P is expected prevalence (considering a prevalence of crowding of 56%), and d is precision (corresponding to effect size, d=0.10).  Considering these paremeters, sample size was 91. We rounded to 100 subjects anticipating possible drop-outs

Reference: Pourhoseingholi MA, Vahedi M, Rahimzadeh M. Sample size calculation in medical studies. Gastroenterol Hepatol Bed Bench. 2013 Winter;6(1):14-7. PMID: 24834239; PMCID: PMC4017493.

The following paragraph has been added to our manuscript, and the corresponding reference has been added too:

“Sample size was calculated using the following formula for cross-sectional studies [18]: n= (Z2 P (1-P)) /d2, where Z represents the statistic corresponding to level of confidence, P represents the expected prevalence, and d represents precision (corresponding to effect size). Based on a confidence level of 95%, a prevalence of crowding of 56%, and a precision of d=0.10, the resulting sample size was 91. In anticipation of possible drop-outs, we rounded up the number of subjects to 100”.

Referee’s comment 3: Please illustrate the measurement points using schematic diagrams or photographs in dental model analysis in Methods section.

Author’s response 3: We have added Figure 1 illustrating the measurement points used.

Figure 1 legend: Measures taken in the dental models. A): incisors tooth size; B): available space in the arch, as described by Sharma and Brown. C): Arch length: anterior (A), posterior (P) and total (A+P) arch length; D): Intercanine (C) and intermolar (M) widths

Referee’s comment 4: Authors should provide a table of correlation coefficients for the measures.

Author’s response 4: The tables with all the correlation coefficients have been added as supplementary material due to their large size. The correlation coefficients have been added as follows:

“Maxillary arch: we found a positive and moderate correlation between TS-ALD and available space (r= 0.373; p<0.01), available posterior space (right side: r= 0.566; p<0.001; left side: r= 0.470; p<0.001), total arch length (r= 0.241; p=0.031), anterior arch length (r= 0.315; p=0.004), and intermolar width (r= 0.325; p=0.003). The variables regarding dental size showed a negative correlation (r= -0.450; p<0.001 for right posterior teeth; r= -0.362; p=0.001 for left posterior teeth, and r= -0.281; p=0.012 for anterior teeth). See Supplementary Table 1.

Mandibular arch: we found a positive and moderate correlation between TS-ALD and available posterior space (right side: r= 0.419; p=0.003; left side: r= 0.380; p=0.008), and a negative correlation with dental size (r= -0.554; p<0.001 for right posterior teeth; r= -0.555; p<0.001 for left posterior teeth, and r= -0.293; p=0.043 for anterior teeth) and mandibular plane (r=-0.287; p=0.048). See Supplementary Table 2.”

Referee’s comment 5: Although the aim of author’s study was to determine whether dental crowding is associated with posterior tooth eruption sequences, authors described that the t-test showed non-significant different for all outcomes in Results section. Therefore, authors should reconsider the purpose of the study (e.g. factors associated with crowding) and add items to be statistically processed.

Author’s response 5: With due respect, we do believe that reporting non-significant results is as important as reporting statistically significant differences. Therefore, in our humble opinion, the lack of association is also an important finding.

Round 2

Reviewer 3 Report

The comments were satisfactorily addressed, and I anticipate an acceptance of the revised manuscript.